# The Fragile Truth of Saliency: Improving LLM Input Attribution via Attention Bias Optimization

**Yihua Zhang**    **Changsheng Wang**    **Yiwei Chen**
**Chongyu Fan**    **Jinghan Jia**    **Sijia Liu**
Michigan State University

## Abstract

Input saliency aims to quantify the influence of input tokens on the output of large language models (LLMs), which has been widely used for prompt engineering, model interpretability, and behavior attribution. Despite the proliferation of saliency techniques, the field lacks a standardized and rigorous evaluation protocol. In this work, we introduce a stress-testing framework inspired by the needle-in-a-haystack (NIAH) setting to systematically assess the reliability of seven popular input saliency methods. Our evaluation reveals a surprising and critical flaw: existing methods consistently assign non-trivial importance to irrelevant context, and this attribution error worsens as input length increases. To address this issue, we propose a novel saliency method based on *Attention Bias Optimization* (ABO), which explicitly optimizes the attention bias associated with each input token to quantify its causal impact on target token generation. ABO robustly outperforms existing methods by $10 \sim 30\%$ in saliency accuracy across diverse NIAH tasks, maintains effectiveness up to 10K-token prompts, and enables practical applications including zero-shot detoxification, sentiment steering, and reasoning-error correction. Our findings highlight the limitations of prevalent attribution methods and establish ABO as a principled alternative for accurate token attribution.

## 1 Introduction

What makes a particular word in a prompt important to a large language model (LLM)? This question lies at the heart of input saliency analysis, a field that seeks to quantify how much each input token contributes to the model's output [1–8]. As LLMs become increasingly embedded in downstream tasks ranging from coding assistants [9, 10] to open-ended reasoning agents [11], understanding their decision-making process is essential for safety, interpretability, and controllability [12, 13]. Input saliency provides a foundation for prompt engineering by identifying which tokens most influence the model's behavior [14, 15], and it offers researchers a window into the inner workings of attention mechanisms and token interactions [16, 17]. As such, saliency analysis has emerged as a cornerstone in the interpretability toolkit for modern language models [18–20].

Diverse input attribution techniques already exists, spanning perturbation-based masking [7, 21], gradient and interaction methods [8, 12], vector-space decompositions [17, 22], Shapley values [23], and linear approximations [12, 24]. Each line of work claims to reveal the "true" importance of input tokens. Yet the field lacks a rigorous, model-agnostic benchmark for judging whether these scores separate meaningful signal from contextual noise, especially in the long-context prompts that define contemporary LLM use. Without such a benchmark, practitioners risk acting on attribution maps that are noisy, unstable, or flat-out wrong.

39th Conference on Neural Information Processing Systems (NeurIPS 2025).

To close this gap we design a *stress test* inspired by the classic "needle-in-a-haystack" (NIAH) evaluation [25]. We embed a single secret message inside a span of distractor text and ask the model to reproduce that message verbatim. Because the distractors have zero causal influence on the output, the ground-truth saliency distribution must concentrate on the secret message (and the final query). This controlled setting lets us audit attribution methods with surgical precision. Our evaluation of six popular techniques [1–6] uncovers a systemic flaw: every method assigns a substantial fraction of saliency to irrelevant tokens, which grows as the context length increases. The misattribution reaches over 90% of the total score for 10K-token prompts, implying that the interpretability tools most widely used today can be *actively misleading* in long-context scenarios.

Motivated by the observation that reweighting an input token's attention can deterministically sway the model's output when two contradictory messages are present, we recast attribution as an *optimization over attention biases*. ABO injects a small, learnable bias term into each attention head and solves for the minimal intervention that maximizes the probability of a target output token. The resulting bias magnitudes serve as faithful saliency scores: they are causal by construction, differentiable, and inexpensive to compute. Empirically, ABO withstands our NIAH stress test and remains robust up to 10K-token inputs. Beyond evaluation, ABO proves useful for zero-shot detoxification, sentiment steering, input token pruning, and rectifying errors in the reasoning traces. To summarize, our contributions are as follows:

• We introduce a **stress-testing framework** for LLM saliency based on the needle-in-a-haystack task, enabling fine-grained diagnosis under long-context inputs.

• We reveal **systemic reliability failures** in six widely-used attribution methods and show that misattribution to irrelevant tokens can exceed over 90 % in 10 K-token prompts, quantifying the brittleness of existing saliency tools.

• We propose **Attention Bias Optimization (ABO)**, a principled, optimization-based technique that delivers token-level saliency scores with superior causal fidelity and scalability to long-contexts.

• We demonstrate ABO's **broad practical utility** across sentiment control, toxic prompt detoxification, and LLM error correction, underscoring its value for both research and real-world deployment.

## 2   Related Work

**LLM input saliency methods.**   Understanding how large language models (LLMs) attribute generated outputs to their preceding input tokens is crucial for model interpretability, debugging, and effective prompt engineering. Several attribution methods have been proposed to elucidate these relationships, typically categorized into gradient-based, vector-based, and perturbation-based approaches. Gradient-based methods measure token importance via gradients obtained through backpropagation. Examples include gradient norm [26–28], gradient-input production [29–31], Layer-wise Relevance Propagation (LRP-XAI) [32, 33], and Generic Attention Explainability (GAE) [8]. Although these methods have demonstrated their value across diverse NLP models and tasks, they predominantly rely on local linear approximations, potentially accumulating errors in deeply stacked transformer layers. Vector-based methods leverage internal model representations, typically attention weights, to infer token importance, exemplified by Attention Rollout [34], GlobEnc [35], and ALTI [36]. However, these approaches implicitly assume linearity in attention score combinations across layers, limiting their fidelity as model depth and complexity increase. Alternatively, perturbation-based techniques [23, 37, 38] measure token significance by assessing changes in model confidence upon token removal or masking. Representative methods include AtMan [6], which combines perturbation strategies with attention mechanisms specifically optimized for transformer models. The evaluations of existing methods typically involve short-context scenarios with limited causal interactions. Consequently, existing benchmarks neither sufficiently explore attribution accuracy under realistic long-context scenarios nor rigorously stress-test the methods' robustness against irrelevant contextual noise. Our work fills this critical gap by introducing a systematic long-context stress-test framework, providing insights into the limitations of current attribution methodologies when applied to LLMs.

**Controlled generation and behavior steering for LLMs.**   A large body of work seeks to steer language–model behavior at *inference* time via prompt design. Manual and automated prompting techniques have shown striking gains on downstream tasks [39–41], and toolkits such as PROMPTSOURCE, RLPrompt, and AutoPrompt make cue construction more accessible [42–46]. Yet LLMs remain

> *[System Prompt] You are a helpful AI chat bot that answers questions for a user. Keep your response short and direct. Don't present information outside the document below or repeat your findings …*
> *[Long Irrelevant Context] July 2006… imagine you were going to spend the weekend at a friend's house on a little island off the coast of Maine.… (irrelevant text continues)… Quiet is another matter.*
> *[Secret Message] The password to get into the zone of Hagazin is "Yin". [Irrelevant Text Resumes] I realize it seems a bit eccentric to take earplugs on a trip to an island off the coast of Maine. If anywhere should be quiet, that should. But what if the person in the next room snored? … Pen and paper wick ideas. [Query on Secret Message] What is the password to get into the zone of Hagazin? Your answer is:*

Figure 1: **Color-coded needle-in-a-haystack (NIAH) [25] prompt.** System instruction, distractor text, secret message, and final query are visually distinguished. When calibrating an LLM saliency method with this example, *only saliency score assigned to the non-gray regions* (yellow, green, blue) is meaningful; any score allocated to gray tokens reflects incorrect attribution.

brittle: small lexical changes can flip predictions or derail chain-of-thought reasoning [47, 48]. Complementary to prompt engineering, *training-time* interventions, e.g. instruction finetuning [49, 50], RLHF and its variants [51–53], or loss terms that privilege user-marked tokens [54–57], improve overall steerability but require additional data or compute. Parallel work explores lightweight adaptation via low-rank or quantized updates (LoRA, AdaLoRA, QLoRA, TOAST) [58–61], and targeted knowledge editing with ROME, MEMIT, MEND, or REMEDI [62–65]. Our saliency-based Attention Bias Optimization is orthogonal to these approaches: it leaves model weights unchanged, requires no extra supervision, and instead manipulates generation by amplifying or suppressing precisely the tokens that the model itself deems most causal.

## 3 A Stress Test of LLM Input Saliency Methods

While input saliency methods have become ubiquitous tools for interpreting LLM predictions, it remains uncertain whether these attribution methods accurately capture token importance under realistic, long-context scenarios. Despite numerous techniques proposed in the literature, their evaluation often relies on relatively simplistic or short-text scenarios [1–3], leaving their generalizability and reliability largely unchecked. To bridge this gap, this section introduces a novel stress-testing framework that systematically probes the robustness and reliability of existing saliency methods under increasingly challenging input conditions. Our aim is twofold: first, to reveal critical limitations of current attribution techniques when exposed to long-context inputs; second, to motivate the necessity of a fundamentally more robust attribution strategy.

**The needle-in-a-haystack (NIAH) test for saliency assessment.** Inspired by the classic *needle-in-a-haystack* paradigm [25], originally developed to assess the information-retrieval capabilities of LLMs in long-context settings, we adapt this experimental setup explicitly for the evaluation of saliency methods. Our adapted NIAH scenario is carefully constructed to provide an explicit, ground-truth saliency distribution for rigorous evaluation. Specifically, as illustrated in **Fig. 1**, we craft prompts comprising four core components:

• a system prompt to establish the overall context, role, and behavior for the LLM's responses;

• a lengthy distractor passage sampled from diverse, public-domain texts (*e.g.*, news articles, Wikipedia entries), explicitly designed to provide irrelevant contextual noise;

• a short, deliberately constructed embedded secret message, which contain keywords (*e.g.*, "Hagazin" and "Yin" in Fig. 1) are intentionally chosen to be *out-of-vocabulary* words absent from standard LLM pre-training corpora, ensuring the uniqueness and singular salience of this message;

• a precise, targeted query placed at the end of the input, asking questions on the secret message hidden in the context.

Before the evaluation, we first make sure the LLM's answer to the query is correct. In the case above, the answer should be "Yin". Then, we use the saliency method to generate saliency scores to all the input tokens with respect to the target output token (*i.e., "Yin"*). The central evaluation metric employed throughout this analysis is the **Hit Ratio**:

$$\text{Hit Ratio} = \frac{\text{Saliency Score}(\text{system prompt} + \text{secret message} + \text{query})}{\text{Total Saliency}}. \quad (1)$$

Intuitively, a higher Hit Ratio approaching 1.0 indicates that a method effectively isolates truly salient tokens, while lower scores indicate a failure in accurately discriminating signal from noise. To ensure fair comparisons across varying methods (each with potentially different saliency scaling conventions), we normalize all saliency vectors to sum to one prior to aggregation.

**Why is NIAH naturally a better fit for saliency calibration?** Critically, the nature of this task involves pure information retrieval and no inferential reasoning is required. Consequently, the ground-truth saliency distribution should strictly allocate importance to three well-defined input regions: (1) the initial system prompt (if present), (2) the embedded secret message, and (3) the final query. All other input tokens, by experimental construction, are irrelevant and thus should be assigned negligible saliency scores. Furthermore, following the original NIAH setting [25], we systematically vary the length of the distractor text from 500 tokens (contexts) up to 4K tokens (contexts), effectively modulating the signal-to-noise ratio within the input while keeping the informative content fixed. This controlled manipulation of input length provides a straightforward yet rigorous means to evaluate each method's robustness to the increasing complexity of realistic long-context scenarios.

**Examination of LLM input saliency via NIAH.** To ensure comprehensive coverage, we evaluate a broad spectrum of **seven representative saliency methods** widely adopted in the literature. These methods span diverse methodological paradigms including gradient-based (GRADNORM [4], INPUT X GRAD [5]), attention-derived (ATMAN [6]), token distribution-based (TDD [2]), Shapley value-based (TOKENSHAP [1]), and linear modeling (AT2 [3]). As a reference, we also compare these methods with a random guess method (NAIVE), which assigns each input token equal saliency. Detailed algorithmic descriptions and hyperparameter selections for each method are provided in Appx. A to facilitate reproducibility and transparency. Our experiments are primarily conducted on a frozen LLaMA-2-7B [66] backbone to ensure controlled comparisons.

**Fig. 2** presents a comprehensive visualization of Hit Ratio performance as a function of input length. Two critical observations immediately emerge: **First**, there is a steep accuracy collapse at long contexts. All evaluated methods experience a precipitous drop in attribution fidelity once the input exceeds approximately 1K tokens. Alarmingly, at input lengths beyond 2K tokens, several methods approach random performance (represented by the gray dashed baseline), highlighting their severe inability to cope with realistic, noisy, long-text inputs. **Second**, there is non-trivial misattribution even at short contexts. Surprisingly, even at the shortest tested context (100 tokens), over half of the attribution mass is consistently misallocated to irrelevant tokens. This persistent misattribution underscores that the root cause is not merely extreme input length, but a fundamental vulnerability in existing methods to structural or positional biases.

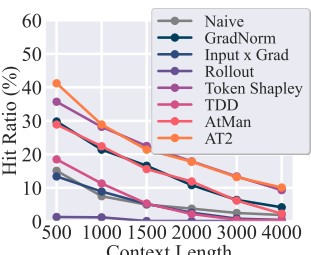

Figure 2: Hit ratio of existing saliency methods on the NIAH stress testing with different input length. Hit Ratio is the fraction of saliency scores assigned to the *true* causal tokens.

To understand the underlying causes of misattribution, **Fig. 3** provides a qualitative heatmap of saliency distributions for representative examples. Strikingly, misattribution does not manifest as

> **[System Prompt]** _You _are _a _helpful _A I _chat _bot _that _answers _questions _for _a _user . _Keep _your _response _short _and _direct . _Don 't _present _information _outside _the _document _below _or _repeat _your _find ings ...
> **[Long Irrelevant Context]** _July _2 0 0 6 ... _imagine _you _were _going _to _spend _the _week end _at _a _friend 's _house _on _a _little _island _off _the _coast _of _Maine ... _Qu iet _is _another _matter .
> **[Secret Message]** _The _password _to _get _into _the _zone _of _H ag az in _is " _Yin " .
> **[Irrelevant Text Resumes]** _I _realize _it _seems _a _bit _ecc ent ric _to _take _ear pl ugs _on _a _trip _to _an _island _off _the _coast _of _Maine . _If _anywhere _should _be _quiet , _that _should . _But _what _if _the _person _in _the _next _room _sn ored _? ... _Pen _and _paper _w ick _ideas .
> **[Query on Secret Message]** _What _is _the _password _to _get _into _the _zone _of _H ag az in _? _Your _answer _is :

Figure 3: Token-level saliency heat-map for a NIAH prompt obtained using AT2 with the context length of $1K$. Each word is background-shaded according to the saliency score (darker=higher). The map reveals substantial saliency weight spilled onto distractor passage, notably high-frequency function words and punctuation.

> *[System Prompt] You are a helpful . . . or repeat your findings . . . [Long Irrelevant Context] July 2006. . . imagine you . . . (irrelevant text continues). . . Quiet is another matter. [Secret Message 1] [Irrelevant Text Resumes] I realize it seems a bit eccentric . . . (irrelevant text continues) . . . Pen and paper wick ideas. [Secret Message 2] [Irrelevant Text Resumes] I hardly ever go back and read stuff . . . (irrelevant text continues) . . . way of having the next. [Secret Message 3] [Irrelevant Text Resumes] The best notebooks I've found . . . (irrelevant text continues) . . . out of space, like a Latin inscription. [Query on Secret Message] What is the password to get into the zone of Hagazin? Your answer is:*

Figure 4: Contradictory NIAH prompt used for the attention-bias study. The long input is color-segmented into a system instruction, alternating blocks of irrelevant context, and three mutually conflicting secret messages, message 1, message 2, and message 3, followed by the final query. The LLM chooses only one secret message to answer the query.

uniform randomness; instead, existing methods disproportionately latch onto structurally salient yet semantically irrelevant tokens, such as high-frequency function words ("the", "is", "of") or syntactic boundaries (sentence breaks, paragraph separations). This critical observation highlights a fundamental flaw: existing methods systematically confuse tokens that dominate attention structure due to frequent repetition or positional prominence with tokens genuinely contributing causally to the model's output. This exposes a systemic weakness of current input-saliency techniques: they conflate attention structure with causal contribution, and the error amplifies with additional distractors.

## 4 Saliency-Biased Attention for Token Attribution

In Sec. 3, we revealed a critical flaw underlying existing saliency methods: their interpretability severely degrades in long-context scenarios. This limitation naturally raises an important question: *Can we not only diagnose these attribution errors but actively intervene to rectify them?* To address this, we revisit our NIAH paradigm, this time introducing a novel and provocative modification by embedding multiple contradictory messages into the input.

**Contradictory secrets reveal attention bias dynamics.** We begin by embedding three *conflicting* secret messages into the input context: Message 1. *The password to get into the zone of Hagazin is "Yin"*, Message 2. *The password to get into the zone of Hagazin is "Vin"*, Message 3. *The password to get into the zone of Hagazin is "Kin"*. See **Fig. 4** as an illustration. At the conclusion of this deliberately confusing passage, we pose the query: *"What is the password to get into the zone of Hagazin?"* Interestingly, we observe a pronounced positional bias in the model's behavior: regardless of their semantic content, LLMs consistently favor the first encountered secret message. This phenomenon suggests that token saliency, model attention toward particular input tokens, may be implicitly influenced by their position rather than their semantic content alone.

**Using attention bias to manipulate LLM's behaviors.** Inspired by this intriguing observation, we probe deeper by deliberately manipulating the model's attention logits, *i.e.*, the intermediate values computed before the softmax normalization in the attention mechanism. Recall that in a standard transformer, the attention score (also called *attention logits*) between a query $\mathbf{q}_i$ and a key $\mathbf{k}_j$ is computed as: $L_{ij} = \dfrac{\mathbf{q}_i \cdot \mathbf{k}_j}{\sqrt{d_k}}$, where $d_k$ is the key dimensionality. These logits are then passed through a softmax to yield attention weights. In order to investigate how attention scores might influence LLM's outputs, we inject an artificially positive attention bias toward tokens in the *second secret message*, gradually increasing its magnitude: $L_{ij} \leftarrow L_{ij} + \alpha_j$, where $[j]$ represents the input token IDs designed to be enhanced with this bias. **Fig. 5** vividly illustrates this experiment: as the bias towards the second secret grows, the probability of outputting the second message's password ("Vin") consistently increases, while the probability associated with the first message's password

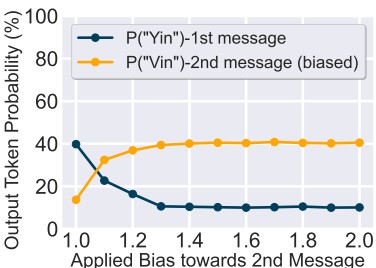

Figure 5: LLM's prediction can be shifted by biasing attention scores. Curves track the probability that the LLM will output the first-secret token "Yin" *vs.* the second-secret token "Vin", when a positive attention bias is gradually injected toward the second secret message. Increasing the bias drives $P(\text{Vin})$ upward from $\sim 14\%$ to $\sim 40\%$, while $P(\text{Yin})$ falls symmetrically, and the two curves cross once the model's final answer flips from the first to the second.

("Yin") correspondingly decreases. Ultimately, the model's final prediction shifts from "Yin" to "Vin". This simple yet profound experiment reveals a deeper insight: **attention logits directly govern token saliency and thus strongly influence model outputs**. Consequently, attention biases can serve as a powerful lever for interpreting and controlling LLM behavior.

**From manipulation to attribution: Attention bias optimization (ABO).** Motivated by the above experiment, we propose ABO, a novel attribution framework. The central insight is straightforward yet compelling: if enhancing a token's attention logits reliably boosts the model's probability of generating a target output, it logically follows that the token is intrinsically salient to that output. Leveraging this principle, ABO reframes token attribution as a differentiable optimization problem over attention logits, thus providing a smooth and precise alternative to discrete masking or traditional gradient-based attribution approaches.

Formally, given an input sequence $\mathbf{x} = (x_1, x_2, \ldots, x_L)$ of length $L$ and a target model-generated output token $y_{\text{target}}$, we introduce learnable attention biases. Specifically, we assign each input token $x_i$ a scalar bias parameter $\alpha_i$, shared across all attention heads and layers. After injecting these biases, we apply standard softmax normalization to yield final attention weights. This soft re-weighting ensures that no token is completely discarded, thus preserving the integrity of the full context. To derive meaningful, interpretable saliency scores, we optimize the parameters $\{\alpha_i\}$ by maximizing the biased model's probability of producing the target token $y_{\text{target}}$:

$$\mathcal{L}(\boldsymbol{\alpha}) = -\log p_{\text{bias}}\big(y_{\text{target}} \mid \mathbf{x}, \boldsymbol{\alpha}\big) + \lambda \sum_{i=1}^{L} \sigma(\alpha_i). \tag{2}$$

Here, the first term ensures tokens that genuinely contribute to generating $y_{\text{target}}$ receive greater saliency. The second term is a carefully designed regularizer: by applying a sigmoid transformation to the biases, it penalizes excessive saliency across irrelevant tokens and promotes sparsity. This choice ensures numerical stability during optimization and yields a sparse, interpretable saliency distribution that highlights genuinely influential tokens. Unlike discrete token masking, ABO never removes tokens but instead smoothly reallocates attention weights across the entire input sequence. This soft, continuous approach provides clearer interpretability: increased bias for a token directly translates to increased model attention and influence. Furthermore, ABO addresses key drawbacks of previous methods, such as instability in gradient-based attribution and arbitrary saliency leakage, by directly coupling token saliency to the model's internal attention dynamics. Consequently, it consistently provides interpretable and robust attributions, even across extremely long contexts.

**Isolating the learned bias yields a clearer and more causal attribution signal.** An important design choice in ABO lies in measuring attribution through the optimized attention bias rather than the sum of the original attention value and bias. While both components jointly influence the model's output during inference, isolating the learned bias yields a cleaner and more causally interpretable signal. Original attention distributions span heterogeneous heads and layers, each capturing structural or positional regularities that may not directly reflect causal relevance. Aggregating them with the optimized bias would inevitably entangle genuine influence with such confounding patterns, thereby weakening attribution clarity. In contrast, the learned attention bias vector forms a unified and end-to-end optimized space that quantifies the minimal perturbation required for a token to affect the model's output. This separation allows ABO to measure causal sensitivity in a stable and comparable manner across contexts, avoiding complex normalization or scale mismatches that arise from mixing heterogeneous attention values.

**Bias sharing across layers and heads stabilizes optimization and improves practicality.** Another design decision concerns sharing the bias parameters across all attention heads and layers. Early variants of ABO explored assigning independent bias parameters to each head and layer, but this additional flexibility did not improve attribution fidelity. Instead, it introduced substantial optimization instability, as interactions among layers amplified local gradient noise and slowed convergence. Empirically, the shared-parameter formulation achieved more stable optimization dynamics and delivered consistently higher saliency precision under equal or lower computational cost. Beyond performance considerations, parameter sharing also enhances the method's practicality, reducing memory footprint, simplifying implementation, and enabling ABO to serve as a lightweight plug-in interpretability module applicable to diverse transformer architectures.

Table 1: Hit Ratio on the single output token NIAH stress test across different LLMs and context lengths. Each cell reports the percentage of saliency mass correctly assigned to the true causal tokens (system prompt + secret message + query), averaged over multiple secret-message insertion depths (from 10% to 90%) within each context and normalized to sum to 100. Higher is better.

| Method | LLaMA2-7B [66] | | | | Mistral-7B-Instruct-v0.3 [67] | | | | DeepSeek-R1-Distilled-QWen-14B [68] | | | |
|---|---|---|---|---|---|---|---|---|---|---|---|---|
| | 1K | 2K | 3K | 4K | 1K | 3K | 5K | 10K | 1K | 3K | 5K | 10K |
| NAIVE | 7.50 | 3.75 | 2.50 | 1.88 | 7.20 | 2.40 | 1.44 | 0.72 | 8.20 | 2.73 | 1.64 | 0.82 |
| GRADNORM [4] | 11.49 | 8.21 | 4.44 | 2.28 | 10.22 | 3.23 | 2.41 | 1.17 | 9.35 | 2.91 | 2.32 | 1.43 |
| INPUT x GRAD [5] | 12.20 | 4.44 | 3.49 | 1.68 | 13.58 | 3.31 | 2.28 | 1.81 | 10.27 | 2.96 | 2.56 | 1.45 |
| TOKENSHAP [1] | 17.73 | 14.82 | 9.91 | 5.24 | 15.54 | 10.27 | 6.37 | 3.73 | 16.73 | 11.43 | 5.59 | 3.81 |
| TDD [2] | 15.57 | 10.83 | 7.29 | 3.37 | 13.28 | 9.23 | 5.51 | 4.39 | 12.71 | 7.68 | 4.83 | 3.51 |
| ATMAN [6] | 19.94 | 16.69 | 12.67 | 4.34 | 20.22 | 13.33 | 7.18 | 3.12 | 18.88 | 12.48 | 6.54 | 2.83 |
| AT2 [3] | 21.15 | 18.27 | 12.28 | 6.62 | 20.43 | 14.41 | 10.03 | 5.59 | 21.76 | 13.65 | 9.86 | 4.99 |
| ABO | 41.75 | 31.72 | 25.54 | 18.83 | 45.34 | 27.77 | 18.92 | 15.43 | 38.89 | 25.14 | 17.37 | 15.44 |

## 5  Experiments

To systematically validate the accuracy, robustness, and utility of our proposed ABO method, we conduct systematic and comprehensive experimental evaluations. The first part (Sec. 5.1) focuses on quantitatively comparing the attribution accuracy of ABO against several strong baselines in controlled scenarios, including single-token and multi-token NIAH stress tests, as well as an input token pruning evaluation in the long-context setting. The second part explores broader applications of ABO, illustrating how accurate saliency attribution can facilitate model behavior manipulation (Sec. 5.2 and Sec. 5.3), and error correction on LLM's input on reasoning tasks (Sec. 5.4).

### 5.1  Saliency Attribution Accuracy

**Single-token NIAH attribution analysis.** First, we evaluate methods using the previously described single-token NIAH setup, measuring their Hit Ratio, *i.e.*, the fraction of saliency correctly attributed to truly causal tokens. As summarized in **Tab. 1**, ABO consistently achieves substantial improvements over all baselines across different models and context lengths. Crucially, ABO maintains a robust Hit Ratio of approximately $15 \sim 45\%$, even as input length expands from 1K to 10K tokens, while the strongest baseline (TOKENSHAP & AT2) rapidly deteriorates to below $5\%$. This stark contrast underscores ABO's unique resilience in handling long-context noise environments. Additionally, ABO's superior performance generalizes robustly across architectures, indicating that its advantage stems fundamentally from its principled attention bias optimization mechanism, rather than model-specific artifacts.

**Multi-token NIAH attribution analysis.** Second, to challenge the saliency methods further, we extend the NIAH setting to secret messages containing multiple target tokens, requiring methods to correctly distribute saliency across multiple output tokens. Results presented in **Tab. 2** affirm ABO's clear superiority: although moving from single-token to multi-token attribution naturally reduces Hit Ratios slightly for all methods, ABO experiences the mildest performance degradation (around $4 \sim 5\%$), maintaining approximately $13 \sim 40\%$ accuracy. In contrast, leading baselines suffer a more substantial drop towards below $10\%$, highlighting ABO's superior capability to capture meaningful attribution across multiple output tokens simultaneously. The persistent robustness of ABO even under increased complexity strongly advocates for its suitability in realistic applications, where multiple tokens frequently contribute jointly to model decisions.

**Long-context token pruning evaluation.** We further validate saliency accuracy in practical scenarios by conducting input token pruning experiments on the challenging LongBench-2wikimqa_e dataset [69]. Tokens are pruned in ascending order of estimated saliency, with QA accuracy measured at progressively higher pruning ratios (see **Tab. 3**). Remarkably, ABO outperforms all baselines, especially under aggressive pruning conditions. For example, at a 95% pruning ratio (retaining only 5% of tokens), ABO achieves a QA accuracy of 51.2% on QWen3-14B, significantly higher than the best baseline's 36.2%. Furthermore, ABO exhibits a flatter accuracy degradation curve across all models tested, demonstrating a stronger ability to reliably identify essential context information. These results confirm ABO's practical value: not only does it accurately reflect token importance,

Table 2: Hit Ratio on the NIAH stress test across with mutiple output tokens different LLMs and context lengths. Other settings follow Tab. 1.

| Method | LLaMA2-7B [66] | | | | Mistral-7B-Instruct-v0.3 [67] | | | | DeepSeek-R1-Distilled-QWen-14B [68] | | | |
|---|---|---|---|---|---|---|---|---|---|---|---|---|
| | 1K | 2K | 3K | 4K | 1K | 3K | 5K | 10K | 1K | 3K | 5K | 10K |
| NAIVE | 7.50 | 3.75 | 2.50 | 1.88 | 7.20 | 2.40 | 1.44 | 0.72 | 8.20 | 2.73 | 1.64 | 0.82 |
| GRADNORM [4] | 8.39 | 6.04 | 3.70 | 2.12 | 9.28 | 3.11 | 2.20 | 1.75 | 8.87 | 2.69 | 1.89 | 1.11 |
| INPUT X GRAD [5] | 8.73 | 5.45 | 3.51 | 2.03 | 8.85 | 2.98 | 1.92 | 1.56 | 9.17 | 2.82 | 2.15 | 1.15 |
| TOKENSHAP [1] | 12.13 | 9.91 | 4.12 | 3.98 | 12.25 | 10.38 | 5.73 | 3.31 | 11.42 | 5.51 | 4.11 | 2.35 |
| TDD [2] | 10.89 | 8.73 | 4.31 | 3.29 | 9.87 | 8.82 | 3.89 | 2.17 | 9.91 | 4.16 | 3.06 | 2.05 |
| ATMAN [6] | 13.27 | 10.29 | 4.98 | 3.71 | 15.55 | 12.27 | 7.73 | 5.41 | 10.40 | 4.41 | 3.71 | 1.51 |
| AT2 [3] | 15.37 | 12.48 | 5.49 | 4.41 | 13.39 | 10.12 | 5.32 | 3.19 | 16.72 | 6.75 | 4.99 | 2.86 |
| ABO | **38.32** | **28.99** | **21.45** | **14.49** | **40.93** | **24.49** | **14.37** | **13.18** | **34.79** | **21.19** | **15.45** | **13.61** |

but it also provides actionable insights enabling substantial runtime efficiency by safely pruning non-essential tokens.

Table 3: QA accuracy (%) on the `LongBench-2wikimqa_e` multi-choice task after pruning input tokens based on saliency scores. Each column indicates the pruning ratio.

| Method | QWen3-8B | | | | | Mistral-Small-24B-Instruct-2501 | | | | | QWen3-14B | | | | |
|---|---|---|---|---|---|---|---|---|---|---|---|---|---|---|---|
| | 10% | 50% | 90% | 95% | 97% | 10% | 50% | 90% | 95% | 97% | 10% | 50% | 90% | 95% | 97% |
| NAIVE | 48.0 | 39.5 | 5.3 | 0.6 | 0.0 | 54.4 | 42.8 | 4.3 | 0.0 | 0.0 | 59.0 | 45.5 | 3.6 | 0.0 | 0.0 |
| GRADNORM | 58.6 | 56.9 | 35.4 | 8.9 | 3.2 | 66.0 | 65.3 | 38.1 | 9.4 | 3.6 | 80.5 | 79.6 | 41.0 | 10.1 | 4.2 |
| INPUT X GRAD | 58.4 | 56.7 | 36.0 | 8.5 | 2.8 | 66.9 | 65.1 | 38.5 | 9.0 | 3.3 | 80.2 | 79.3 | 41.2 | 9.8 | 3.8 |
| TDD | 58.5 | 55.6 | 33.2 | 7.2 | 2.4 | 67.6 | 63.9 | 36.8 | 8.3 | 2.9 | 78.4 | 77.3 | 39.1 | 9.1 | 3.5 |
| ATMAN | 59.1 | 58.6 | 44.8 | 28.2 | 5.2 | 67.1 | 66.5 | 49.5 | 31.3 | 8.3 | 81.8 | 80.9 | 54.1 | 34.1 | 9.1 |
| AT2 | 59.4 | 58.9 | 46.3 | 30.0 | 7.8 | 67.4 | 66.9 | 51.0 | 32.5 | 8.7 | 82.1 | 81.2 | 56.0 | 36.2 | 9.5 |
| Ours | **59.3** | **59.0** | **52.6** | **40.3** | **9.9** | **67.5** | **67.1** | **58.7** | **47.0** | **12.2** | **82.3** | **81.7** | **65.1** | **51.2** | **15.6** |

## 5.2 LLM Saliency for Zero-Shot Toxicity Suppression.

We further demonstrate the practical utility of accurate saliency attribution through the task of zero-shot toxic language suppression. Despite recent advancements, LLMs remain prone to generating toxic outputs in response to certain inputs [70, 71]. Unlike prior reactive approaches aimed at mitigating toxicity post-generation [72–74, 74], our approach proactively identifies and removes toxic triggers directly from the prompt, ensuring safer downstream generation.

Table 4: Toxicity suppression results on REALTOXICPROMPTS [72] using Mistral-7B-Instruct-v0.3 [67]. Scores on six toxicity attributes provided by the Perspective API; lower values indicate better suppression.

| Method | Toxicity | Severe Toxicity | Sexually Explicit | Threat | Profanity | Identify Attack |
|---|---|---|---|---|---|---|
| Original | 0.53 | 0.24 | 0.37 | 0.23 | 0.42 | 0.19 |
| TDD | 0.36 | 0.21 | 0.29 | 0.18 | 0.33 | 0.14 |
| TOKENSHAP | 0.41 | 0.18 | 0.17 | 0.14 | 0.26 | 0.12 |
| AT2 | 0.28 | 0.15 | 0.21 | 0.11 | 0.14 | 0.09 |
| ABO | **0.24** | **0.12** | **0.14** | **0.08** | **0.07** | **0.05** |

**Experiment setup.** Specifically, we utilize the REALTOXICPROMPTS [72] dataset, comprising 1,225 prompts known for eliciting highly toxic outputs, and evaluate our method using the Mistral-7B-Instruct-v0.3 model [67]. Given our rigorous assessment of saliency attribution methods in the previous subsection, we selectively benchmark against the top-performing saliency approaches, namely TOKENSHAP [1], AT2 [3], and TDD [2]. During the experiment, we first designate the predefined toxic words from WORDFILTER [72] as target tokens, identifying critical tokens (toxic triggers) via saliency scores, which we subsequently neutralize by replacing them with meaningless space tokens. Following the procedure of prior work [73, 75], we limit generations to 20 tokens and evaluate the resulting toxicity using six toxicity attributes provided by Perspective API, ensuring a clear and rigorous comparison of saliency-based toxicity suppression efficacy among our chosen baseline methods.

**Results and analysis.** As shown in **Tab. 4**, our method achieves the lowest toxicity scores across all six attributes, substantially outperforming all baselines. Compared to the original outputs, ABO

Table 5: Sentiment steering results on OPENWEBTEXT [76] using Mistral-7B-Instruct-v0.3 [67]. Generated outputs are classified via Huggingface sentiment analysis [77]; higher indicate stronger steering efficacy.

| Method | Neutral → Negative Negative Ratio (%) | Neutral → Positive Positive Ratio (%) |
|---|---|---|
| Original | 53.6 | 46.4 |
| TDD | 75.4 | 59.9 |
| TOKENSHAP | 80.2 | 68.3 |
| AT2 | 69.3 | 66.7 |
| ABO | **89.7** | **81.5** |

reduces overall toxicity by more than half, and consistently yields the strongest suppression in severe categories such as *threat*, *profanity*, and *identity attack*. While AT2 emerges as the strongest baseline, it still lags behind across every metric. These results confirm that our saliency estimates more accurately pinpoint true toxic triggers, enabling more effective prompt sanitization and offering a robust foundation for safe LLM deployment.

## 5.3 LLM Saliency for Sentiment Steering

We next demonstrate ABO's applicability in zero-shot sentiment steering, an essential task for safely and reliably guiding LLM-generated content towards a desired sentiment polarity.

**Experiment setup.** To rigorously assess our approach, we use 5,000 neutral prompts sourced from the OPENWEBTEXT [76] corpus to feed the Mistral-7B-Instruct-v0.3 model [67]. Specifically, we designate negative sentiment words from SENTICNET [78] as target tokens and positive words as alternative tokens when steering towards positive sentiment, and reverse this assignment when steering negatively. Subsequently, we utilize saliency attribution methods, focusing exclusively on the highest-performing methods identified previously: TOKENSHAP [1], AT2 [3], and ATMAN [6] to pinpoint the single most salient sentiment cue within each prompt. This identified token is then directly replaced by the explicit keyword (either "positive" or "negative" accordingly) to guide the model's sentiment. Following prior work [79], each prompt undergoes exactly one token replacement due to their relatively short lengths. For evaluation, we exclusively report the sentiment distributions of generated outputs using the Huggingface sentiment analysis classifier [77], clearly reflecting the efficacy of each method in steering sentiment polarity.

**Results and analysis.** As shown in **Tab. 5**, a single ABO-guided token edit shifts *89.7%* of neutral prompts to negative and *81.5%* to positive sentiment, outperforming the strongest baseline (TOKENSHAP: 80.2%/68.3%). The gap widens in the harder "neutral→positive" scenario, highlighting ABO's ability to pinpoint the *single* most causal sentiment cue. Together, these results confirm that bias optimization yields sharper, more actionable saliency than existing heuristics, enabling reliable zero-shot sentiment control with minimal input modification.

---

[**Question**] In a dance class of 20 students, 20% enrolled in contemporary dance, 25% of the `remaining` enrolled in jazz dance, and the rest enrolled in hip-hop dance. What percentage of the students enrolled in hip-hop dance?

---

[**Reasoning (Incorrect by Default)**] <think> To determine the percentage of students enrolled in hip-hop dance, let's break down the information provided: ... Students enrolled in Jazz Dance: ( 25% ) of 20 students \[ ~~0.25 × 20 = 5~~ students \] ... <think>

Final Answer: \(\boxed{55\%}\)

---

[**Reasoning (Correct with Attention Bias)**] <think> To determine the percentage of ... Students enrolled in Jazz Dance: ( 25% ) of the **remaining** students after contemporary dance \[ 20 - 4 = 16 students \] Then, 25% of 16: \[ **0.25 × 16 = 4** students \]... <think>

Final Answer: \(\boxed{60\%}\)

---

Figure 6: Attention bias fixes a reasoning error. Highlighting the token "remaining" (yellow) and adding a positive bias to its attention logits redirects the model to the correct interpretation, changing the wrong 55% answer (left, red) to the correct 60% answer (right, green).

### 5.4 LLM Error Correction via ABO

Beyond interpretability and behavior steering, our ABO (attention bias optimization) approach provides a novel pathway for error correction during reasoning tasks performed by LLMs. Despite their impressive reasoning capabilities, even advanced reasoning-centric models such as DeepSeek-R1-Distilled-QWen-7B [68] occasionally commit errors due to misreading critical elements within problem statements. To illustrate this, consider a simple mathematical question from the GSM8K dataset [80] depicted in **Fig. 6**. Initially, the model erroneously overlooks the keyword *"remaining"*, consequently misunderstanding the problem statement and generating an incorrect answer. However, by explicitly introducing a positive attention bias to the token corresponding to the overlooked keyword, we effectively realign the model's attention, enabling the model to correctly interpret the question and subsequently correct its output.

Motivated by this observed phenomenon, we systematically examine a broader set of examples to quantitatively assess the efficacy of our attention-based error correction strategy. Specifically, we identify 50 distinct math reasoning problems from the GSM8K dataset initially answered incorrectly due to misreading or overlooking crucial textual cues. For each problem, we manually select and positively bias the attention logits corresponding to the critical tokens identified as frequently misinterpreted. Remarkably, after applying such targeted attention biases, 45 out of these 50 problems (90%) were subsequently answered correctly without any modifications to the input text itself. This result strongly indicates that the errors were indeed a consequence of attentional oversight rather than inherent model deficiencies. See Appx. B for more examples.

## 6 Limitation and Conclusion

We revisited *input saliency* for LLMs and showed that six popular attribution methods collapse in long-context settings, misplacing most of their mass on distractors. Our NIAH (needle-in-a-haystack) stress test makes this failure mode transparent and measurable. Building on the empirical insight that tiny shifts in attention logits can deterministically flip a model's choice among contradictory messages, we introduced ABO, which frames attribution as a lightweight per-token bias optimization on attention logits. We empirically demonstrated the accuracy of ABO's saliency scores and its ability in zero-shot detoxification, sentiment steering, and correction of reasoning errors.

Despite these promising results, this work also reveals a more fundamental limitation of current *input saliency* research for LLMs. Our evaluation shows that the gap between how saliency methods are *perceived* to work and how they actually behave in practice remains surprisingly large. While ABO substantially narrows this gap relative to prior methods, its absolute performance still leaves ample room for improvement. This persistent discrepancy stems from the inherent scale and complexity of modern LLMs, where attribution noise, token interactions, and non-linear dependencies grow rapidly with model size. Consequently, even small estimation errors in saliency can accumulate and distort the interpretation, posing a challenge that goes beyond the reach of any single method.

From an algorithmic perspective, ABO's per-example optimization introduces additional variance across runs and examples, especially when the target likelihood landscape is highly non-convex. This sensitivity makes results less stable in certain contexts and increases computational cost for large-scale deployment. Future work could explore amortizing bias inference through meta-learning or distillation-based techniques, allowing approximate but faster and more consistent saliency estimation. Furthermore, our current stress test mainly targets retrieval-style prompts; extending it to multi-step reasoning, dialogue, and multimodal tasks would provide a broader view of attribution fidelity under real-world conditions. Overall, ABO represents an encouraging step toward causally grounded saliency analysis for large language models. It not only improves attribution fidelity under long-context noise but also exposes deeper methodological gaps in how we interpret complex neural systems, gaps that will require sustained community effort to close.

## Acknowledgment

This work was supported by the National Science Foundation (NSF) CISE Core Program Award IIS-2207052, the NSF Cyber-Physical Systems (CPS) Award CNS-2235231, the NSF CAREER Award IIS-2338068, the ARO Award W911NF2310343, the Cisco Research Award, the Amazon Research Award, and the IBM PhD Fellowship Award.

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

# Appendix

## A   Experiment Setup

All saliency methods were normalized to ensure that their respective attribution scores sum to one.

**Gradient Norm (GRADNORM)** computes saliency scores based on the $\ell_2$ norm of the gradient of the model's output with respect to the input token embeddings. Specifically, the score for each token $x_i$ is calculated by taking the L1 norm of the gradient $\nabla_{x_i} q(y_t|x)$, where $q(y_t|x)$ is the model's predicted probability for token $y_t$ given input $x$.

**Gradient × Input (INPUT×GRAD)** modifies the Gradient Norm approach by calculating the dot product between each input embedding $x_i$ and its gradient $\nabla_{x_i} q(y_t|x)$, thereby emphasizing tokens highly expressed in the model's predictive process.

**Attention Rollout (ROLLOUT)** tracks the flow of attention information through transformer layers by recursively multiplying attention matrices across layers. Given raw attention matrices $A(l_i)$ at each layer $l_i$, attention from higher layers is multiplied downwards layer by layer, creating cumulative attention maps that reflect the propagation of token importance from input layers up to the output layer (`https://github.com/hmohebbi/context_mixing_toolkit`).

**TokenSHAP (SHAP)** estimates Shapley values for tokens via Monte Carlo sampling. The importance of each token is determined by evaluating its marginal contribution to the model's output across randomly sampled subsets of tokens, combining essential combinations with randomly sampled ones to ensure both computational efficiency and accurate estimation (`https://github.com/ronigold/TokenSHAP`).

**Token Distribution Dynamics (TDD)** analyzes token saliency using token distribution projections within the embedding space via the language model head. Variants such as forward, backward, and bidirectional TDD measure token saliency through their distributional shifts and causal contributions to model predictions (`https://github.com/zijian678/TDD`).

**Attention Manipulation (ATMAN)** generates relevance maps through memory-efficient perturbations of transformer attention mechanisms. It applies token-level perturbations directly within the attention score space, reducing computational overhead and facilitating efficient interpretation of model decisions (`https://github.com/Aleph-Alpha/AtMan`).

**Attribution with Attention (AT2)** treats attention weights from individual heads as learnable features. This method uses ablation-based training to learn attention-head coefficients, enabling efficient yet accurate token attribution across different contexts and significantly reducing computational requirements (`https://github.com/MadryLab/AT2`).

**ABO**: For ABO, we optimized the bias parameters $\{\alpha_i\}$ using the Adam optimizer with a typical learning rate of $0.01$, applying a decay schedule after a fixed number of iterations. The optimization was capped at 50 to 200 steps, depending on the context length, with early stopping triggered by the convergence of the target token probability. To promote sparsity and focus in the saliency distribution, we applied a sigmoid-transformed $\ell_1$ regularizer with weight $\lambda$ ranging between $10^{-1}$ and $1$. All bias parameters were initialized randomly to avoid introducing inductive bias at initialization.

### A.1   Compute Environment

All experiments were conducted exclusively for evaluation purposes, with no model training or fine-tuning involved. We utilized NVIDIA H100 GPUs for all runs, and the total compute consumption was approximately 1000 GPU hours. This estimate includes forward passes, saliency score extraction, attention bias optimization, and repeated runs for multi-seed robustness analysis across multiple saliency methods and model backbones.

## B   Additional Experiment Results

In this section, we provide additional qualitative examples (Fig. A1∼Fig. A5) to illustrate how BPO can be used to correct model reasoning errors by redirecting attention to overlooked yet critical tokens. These examples are drawn from the GSM8K dataset and demonstrate instances where the

[**Question**] Paul went to a shop to buy some groceries. He bought some bread for 2, butter for 3, and juice for two times the price of the  bread . He had 15 for his shopping. How much money did Paul have left?

[**Reasoning (Incorrect by Default)**] \<think\> Let's determine how much money Paul has left after his shopping step by step. The bread costs 2. ... The juice costs two times the price of the ~~butter~~ ... \<think\>
Final Answer: \(\boxed4\)

[**Reasoning (Correct with Attention Bias)**]\<think\> Let's determine how much money Paul has left after his shopping step by step. The bread costs 2. ... The juice costs two times the price of the **bread** ... \<think\>
Final Answer: \(\boxed6\)

Figure A1: **Example 1.** Initially, the model misidentifies "juice costs twice the butter" instead of "twice the bread". With attention bias on the word "bread," the model corrects its interpretation and computes the correct remaining balance.

[**Question**] There are some jelly beans in a jar. Three fourths of the  red  jelly beans are red, and one quarter of the jelly beans are coconut flavored. If 750 jelly beans are coconut flavored, how many jelly beans are there in the jar?

[**Reasoning (Incorrect by Default)**] \<think\> Let's solve the problem step by step. Given: ... Since one quarter of the ~~total jelly beans~~ equals 750, I can set up the equation: ... \<think\> Final Answer: \(\boxed3000\)

[**Reasoning (Correct with Attention Bias)**]\<think\> ... That means if I have a total number of jelly beans, say, T, then **the number of red jelly beans** would be (3/4) * T. ... \<think\> Final Answer: \(\boxed4000\)

Figure A2: **Example 2.** The model initially misattributes the fraction to total jelly beans. Highlighting "red" refocuses attention on the subject of the fraction, enabling correct inference of the jar size.

model initially produced incorrect answers due to misinterpretation. By applying positive bias to the appropriate input tokens, we successfully guide the model to revise its understanding and produce the correct answers—without modifying the input text. Each example shows the original faulty reasoning on the left and the corrected reasoning with attention bias on the right. These results showcase the potential of attention-level interventions for LLM robustness and post hoc debugging.

# C   Broader Impact

This work advances the field of trustworthy AI by proposing Attention Bias Optimization (ABO), a principled method for accurate token attribution in large language models (LLMs). By identifying and correcting systemic flaws in six widely-used saliency methods, our work enables more faithful, scalable, and robust interpretability, especially in long-context settings.

From an ethical standpoint, ABO enhances transparency in LLM behavior, reducing the risk of misinterpretation in high-stakes applications such as legal analysis, education, and healthcare. However, accurate attribution also raises dual-use concerns: malicious actors could leverage improved interpretability to reverse-engineer sensitive prompts or fine-tune manipulative behavior. Future research should explore safeguards that balance interpretability with misuse prevention.

At the societal level, our method supports the development of safer AI systems by enabling applications such as zero-shot toxicity suppression, error correction, and sentiment steering. These capabilities can improve the reliability and controllability of AI assistants and mitigate harms caused by misaligned or biased outputs. Still, we caution that attribution is not equivalent to accountability; robust evaluation and user education remain necessary.

**[Question]** A company pays each of its employees 600 in a `month` . The company has a policy of increasing the salaries of each of its employees by 10% of the initial salary every year for those who've stayed in the company for five years. If Sylvie just clocked 5 years in the company last December, what's her annual salary after three more years of service?

[**Reasoning (Incorrect by Default)**] \<think\> Alright, so I have this problem about Sylvie's salary ... First, it says that the company pays each employee ~~600 per year~~. ...\<think\>
Final Answer: \\(\boxed780\\)

**[Reasoning (Correct with Attention Bias)]**\<think\> First, I need to determine Sylvie's current monthly salary. Since she has been with the ... Starting salary: **600 per month**. ... \<think\>
Final Answer: \\(\boxed10,541.52\\)

Figure A3: **Example 3.** The model mistakes the salary to be yearly instead of monthly. Biasing the word "month" shifts the model toward the correct interpretation and output.

**[Question]** Brian's friend Bobby has 5 fewer than 3 times as many video games as Brian does. If Brian has 20 video games but lost 5 right `after` the comparison was made, how many does Bobby have?

[**Reasoning (Incorrect by Default)**] \<think\> Okay, let's tackle this problem step by step. ... Determine the current number of video games Brian has: Originally, Brian has 20 video games. He lost 5 video games. ~~Current number of Brian's games= 20 − 5 = 15~~ ... \<think\> Final Answer: \\(\boxed40\\)

**[Reasoning (Correct with Attention Bias)]**\<think\> Alright, let's try to solve this problem step by step. ... The phrase implies that the comparison occurred, and then Brian lost 5. So, it seems like the comparison was done when Brian had 20 games, **and after that, he lost 5**. ...\<think\>
Final Answer: \\(\boxed55\\)

Figure A4: **Example 4.** The model mistakenly adjusts the number of video games after the comparison. Emphasizing the word "after" leads to the correct timing of the event and the correct answer.

We encourage continued efforts to integrate interpretability into broader frameworks of AI alignment, governance, and fairness auditing to ensure that attribution advances benefit diverse populations responsibly.

**[Question]** Jerome had 4 friends who came to visit him on a certain day. The first friend pressed on the doorbell 20 times before Jerome opened, the second friend pressed on the doorbell 1/4 times more than Jerome's first friend. The third friend pressed on the doorbell 10 times more than the `fourth` friend. If the fourth friend pressed on the doorbell 60 times, how many doorbell rings did the doorbell make?

[**Reasoning (Incorrect by Default)**] <think> Alright, let's tackle this problem step by step. ... First Friend: Pressed the doorbell 20 times. Second Friend: Pressed the doorbell 1/4 times more than the first friend. Third Friend: Pressed the doorbell 10 times more than the ~~second~~ friend. Third Friend: 250 times ...<think> Final Answer: \(\boxed605\)

[**Reasoning (Correct with Attention Bias)**] <think> First, identify the number of times each friend pressed the doorbell. ... Fourth Friend: The fourth friend pressed the doorbell 60 times. Third Friend: The third friend pressed the doorbell 10 times **more than the fourth friend**. Calculation: [ 10 × 60 = 600 times ] ... <think> Final Answer: \(\boxed605\)

Figure A5: **Example 5.** The model incorrectly attributes the comparison for the third friend to the second friend. Attention bias on the word "fourth" enables accurate attribution and total computation.

