# OpenReview forum: "The Fragile Truth of Saliency: Improving LLM Input Attribution via Attention Bias Optimization"
_NeurIPS.cc/2025/Conference — NeurIPS 2025 spotlight_

### Official Review · Reviewer_Rxaz · 2025-06-19

**Clarity:** 3
**Significance:** 3
**Originality:** 3
**Rating:** 5
**Confidence:** 4

**Summary:**

This paper explores the degradation of existing saliency methods in long-context scenarios and proposes a solution: Attention Bias Optimization (ABO). The authors conduct extensive experiments to validate ABO's effectiveness and further demonstrate its utility in controllable text generation. Overall, this is an interesting and in-depth study with meaningful contributions.

**Questions:**

Please see Strengths And Weaknesses

**Ethical Concerns:**

["NO or VERY MINOR ethics concerns only"]

**Final Justification:**

The authors have addressed all my concerns, so I raised my score accordingly.

**Limitations:**

yes

**Paper Formatting Concerns:**

The authors should remove the period at the end of the Section 5.2 title.

**Quality:**

3

**Strengths And Weaknesses:**

Strengths:

1.    The observed relationship between attribution quality and input length is both novel and insightful.
2.    The proposed ABO method is simple yet effective.
3.    The paper presents its motivation, methodology, and applications clearly and coherently.

Weakness:

Below are my questions and concerns. I list more than usual because I believe the paper is promising and could be strengthened with further clarifications.

1.	Regarding the statement on line 121: does the LLM achieve 100% accuracy on NIAH queries across all context lengths (from 500 to 10k)?

2.	Lines 156–158 reference "100 tokens," but Figure 2 shows results for 500 tokens. Should this be corrected to 500 tokens?

3.	In the proposed method, is the learned alpha directly used as the final attribution score for each token?

4.	In Figure 5, the output probability appears to be influenced by both the induced bias and the original attention scores (lines 194 and 200). However, the final ABO method uses only the bias scores. Please justify why the original attention scores can be ignored.

5.	For the main experiments (Tables 1 and 2), please provide dataset statistics. Also, how many training samples were used to train ABO?

6.	The experiments in Tables 1 and 2 may not present a fair comparison. ABO is trained on the same data used for testing, while other methods lack access to this prior knowledge. This is more akin to comparing supervised with unsupervised methods, which disadvantages the baselines.

7.	In contrast, the experiment in Table 3 offers a fairer comparison and is more convincing if ABO utilizes the NIAH dataset for training instead of LongBench-2wikimqa_e. May I confirm waht is the training data for this experiment? Also,  I recommend including more experiments of this nature.

8.	Is ABO only applicable in long-context scenarios? Testing it in classic settings (e.g., TDD experiments) would help validate its generalizability.

9.	Another concern is whether ABO truly learns attribution scores or merely copies input repeated tokens. Simple copying can also reduce training loss but may not generalize beyond NIAH scenarios. This distinction needs to be empirically verified. This is also another reason I ask more experiments as questions 7 and 8 above.

10.	In Sections 5.2 and 5.3, many settings and operations for controllable text generation seem directly adopted from the TDD framework. The authors should explicitly cite TDD and clearly state, like “We follow the experimental settings and procedures for controllable text generation as described in TDD.”

---

> ### Author Rebuttal · Authors · 2025-07-31
>
> We greatly appreciate your thoughtful feedback and insightful comments. Your suggestions have been invaluable in helping us improve the clarity and quality of our work. Below, we address each of your points in detail.
>
> **Q1. Regarding the statement on line 121: does the LLM achieve 100% accuracy on NIAH queries across all context lengths (from 500 to 10k)?**
>
> **A1.** Yes, your understanding is correct: for every NIAH example in our evaluation, we ensured that the LLM produced the correct answer to the query before applying any saliency method. This serves as a necessary prerequisite for  evaluating the correctness of saliency under the correct prediction and for conducting additional analysis in the NIAH setup . This is also why in Table 1 and 2, different models may use different max test length, which is the max length that the model makes no mistakes in NIAH prior to further assessing saliency methods.
>
> **Q2. Lines 156–158 reference "100 tokens," but Figure 2 shows results for 500 tokens. Should this be corrected to 500 tokens?**
>
> **A2:** Thanks for your careful reading! You are correct: It should be 500 tokens. We will correct this typo in our revision.
>
> **Q3. In the proposed method, is the learned alpha directly used as the final attribution score for each token?**
>
> **A3.** We apologize for the ambiguity. No, the raw learned alpha values are not directly used as the final attribution scores. Since the alpha values can be both positive and negative, we apply a softmax to convert them. This ensures that strongly negative values are suppressed (near-zero importance), and the attribution scores sum to one, making them interpretable as relative importance percentages across the input tokens. We will clarify this in the revision.
>
> **Q4. In Figure 5, the output probability appears to be influenced by both the induced bias and the original attention scores (lines 194 and 200). However, the final ABO method uses only the bias scores. Please justify why the original attention scores can be ignored.**
>
> **A4.** We appreciate the reviewer’s keen observation, during inference, both the learned attention bias and the original attention scores jointly influence the model’s output. However, when computing attribution scores, we intentionally exclude the original attention scores and rely solely on the optimized bias terms. This design is based on the following practical  considerations.
>
> Separation from confounding attention patterns: Original attention scores are distributed across many heads and layers, each with distinct contextual roles and dynamic ranges. These scores often reflect structural or positional biases rather than causal token influence. Including them directly in attribution would risk entangling meaningful signal with these confounding factors.
> Stability and interpretability of the bias space: The learned attention bias vector is optimized end-to-end in a single, unified space. As such, it provides a clean, direct measure of how much “perturbations” (sensitivity or additional emphasis) a token needs to shift the model's output, offering a more faithful and causally grounded saliency signal. Combining it with the original attention would require complex normalization across layers and heads, introducing potential scale mismatches and reducing attribution clarity.
>
>
> **Q5. For the main experiments (Tables 1 and 2), please provide dataset statistics. Also, how many training samples were used to train ABO?**
>
> **A5.** Thank you. To clarify, ABO does not involve “training”. It is not a method that pretrains or fine-tunes a model using a dataset. Instead, given a frozen model, a specific input prompt, and a target output token, we perform a lightweight optimization over the attention bias vector, which results in the input saliency. This is consistent with other post-hoc input saliency methods, such as gradient-based attribution techniques. While our optimization uses backpropagation, it operates entirely on the given example, i.e., no additional data or training samples are involved. And the dimension of the optimization variables (attention bias vector) is aligned with the input example dimension. Therefore, there is no additional training phase for ABO.
>
> Regarding dataset statistics: As stated in the captions of Tables 1 and 2, our evaluation in the NIAH setup is conducted as follows: for each context length (e.g., 1K–10K tokens), we insert the secret message at 9 different positions, uniformly spanning from 10% to 90% of the prompt. The reported attribution accuracy is the average over these positions, ensuring robustness across varying insertion points.  Apart from this, no additional datasets are involved.
>
> In short, NIAH is introduced as a new evaluation framework to stress-test attribution methods, not as a training set for ABO. ABO itself is a post-hoc input saliency method that operates on given models and significantly outperforms existing input attribution methods across a range of tasks, including NIAH stress testing (Sec. 5.1), toxicity suppression (Sec. 5.2), sentiment steering (Sec. 5.3), and LLM error correction (Sec. 5.4).
>
>
> **Q6. The experiments in Tables 1 and 2 may not present a fair comparison. ABO is trained on the same data used for testing, while other methods lack access to this prior knowledge. This is more akin to comparing supervised with unsupervised methods, which disadvantages the baselines.**
>
> **A6.** This might be a misunderstanding of ABO. As clarified in our previous response ABO does not involve any training or external training dataset. The optimization in ABO is entirely post-hoc, i.e., performed for each test instance given an already-trained model, without requiring any external data, supervision, or prior knowledge. Therefore, ABO is not "trained on the test data," and the comparison remains fair.
>
> **Q7. In contrast, the experiment in Table 3 offers a fairer comparison and is more convincing if ABO utilizes the NIAH dataset for training instead of LongBench-2wikimqa_e. May I confirm what the training data for this experiment is? Also, I recommend including more experiments of this nature.**
>
> **A7.** Thank you for the suggestion. As mentioned in our responses to Q5 and Q6, ABO does not involve any training on datasets such as NIAH or LongBench-2wikimqa_e. NIAH is introduced as a new evaluation framework to stress-test attribution methods, not as a training set for ABO.
>
>
> **Q8. Is ABO only applicable in long-context scenarios? Testing it in classic settings (e.g., TDD experiments) would help validate its generalizability.**
>
> **A8.** Thank you for the suggestion. ABO is not limited to long-context scenarios. In fact, we evaluated its performance across a range of context lengths from 1K to 10K tokens in Tables 1-2, demonstrating consistent effectiveness even in shorter inputs.
>
> We agree that TDD presents several strong examples of attribution evaluation, and we have already incorporated some of its experimental setups, specifically those in Table 4 (toxicity suppression) and Table 5 (sentiment steering). As for the other classic attribution experiments used in TDD, we chose not to adopt them initially due to the lack of clearly defined ground-truth saliency labels, which makes rigorous and objective evaluation challenging.
>
> **Q9. Another concern is whether ABO truly learns attribution scores or merely copies input repeated tokens. Simple copying can also reduce training loss but may not generalize beyond NIAH scenarios. This distinction needs to be empirically verified. This is also another reason I ask more experiments as questions 7 and 8 above.**
>
> **A9.** We appreciate the reviewer’s concern. However, we believe this may stem from a misunderstanding similar to those addressed in Q5–Q7: ABO does not involve training, and therefore the notion of “reducing training loss” does not apply in the NIAH setting. To stress, ABO performs post-hoc optimization for input saliency. There is no learning across examples or fitting to a dataset.
>
> Furthermore, the effectiveness of ABO extends beyond the NIAH setup, as shown in Tables 4 and 5, ABO identifies causal tokens in diverse real-world tasks such as sentiment steering and toxicity suppression, where the key tokens are not duplicated. These results support the claim that ABO meaningfully captures attribution rather than relying on token repetition.
>
> **Q10. In Sections 5.2 and 5.3, many settings and operations for controllable text generation seem directly adopted from the TDD framework. The authors should explicitly cite TDD and clearly state, like “We follow the experimental settings and procedures for controllable text generation as described in TDD.”**
>
> **A10.** Thank you for pointing this out. We appreciate the reminder, and we will make the  citation and clarification more explicit in the revised version.

---

> > ### Comment · Reviewer_Rxaz · 2025-08-01
> >
> > Thank the authors for their clarification. Most of my concerns are well-addressed. I will reflect this on my final scores.

---

> > > ### Author Response · Authors · 2025-08-01
> > > **Thank you!**
> > >
> > > Thank you very much for your thoughtful follow-up and for taking the time to engage with our responses. We truly appreciate your acknowledgment that our clarifications addressed your concerns, and we're grateful for your willingness to reflect this in your final rating. Your comments and suggestions have been invaluable in helping us strengthen both the paper and its broader framing.

---

### Official Review · Reviewer_iMDk · 2025-06-22

**Clarity:** 4
**Significance:** 3
**Originality:** 3
**Rating:** 5
**Confidence:** 4

**Summary:**

The paper assesses the reliability seven popular input saliency evaluation protocols and reaches the conclusion that existing methods assign importance to non-relevant tokens through the needle-in-a-haystack (NIAH) experiment. In addition, the problem is more severe under long input length. Attention Bias Optimization (ABO) is proposed to resolve this problem. ABO learns the attention bias to be injected in each token that maximizes the probability of generating the target output. Higher attention bias indicates higher saliency of the input token.
ABO robustly outperforms existing methods not only in NIAH but also other practical scenarios.

**Questions:**

Why do we measure attention bias instead of original attention value plus attention bias? My concern is that if the attention of a relevant token is already high, the learned bias would turn out to be small, as adding additional bias does not make much impact on the final probability of outputting the  target token. Consider the example in Section 4, Line 178, if I set the target token as the first secret message "Yin", what would be the learned attention bias of "Yin"? What would be the original attention of "Yin"?

**Ethical Concerns:**

["NO or VERY MINOR ethics concerns only"]

**Final Justification:**

I keep my original score of 5. The authors clearly resolve all my concerns.

Why no higher score: The significance of the paper's approach in practice still needs to be test by time despite the three scenarios tested in the paper.

**Limitations:**

Yes

**Paper Formatting Concerns:**

No issues

**Quality:**

4

**Strengths And Weaknesses:**

**Strength in Quality**: Abundant convincing experiments are conducted to support the claim of the paper.  First, the paper investigates the NIAH setup, in which the relevant tokens are well-defined. Therefore, it is persuasive to show the limitations of existing evaluation methods. Second, the paper runs multiple practical experiments such as zero-shot detoxification, sentiment steering, and reasoning-error correction, in order to demonstrate the real-world strength of the proposed ABO method.

**Strength in Clarity**: The paper is well-structured. First, a simple toy experiment, NIAH,  is studied to show limitations of existing methods. Then, the ABO method is proposed to resolve the limitations. Finally, more practical tests are conducted to demonstrate the strength of ABO.

**Strength in Significance**:  The paper contributes to the interpretability of LLMs by proposing a reliable saliency evaluation protocol. The evaluation is computationally efficient because no model training is involved.

**Strength in Originality**: The paper introduces a new norm of evaluating saliency of input tokens that intervenes on generation only instead of model weights.

**Weakness in Significance**: ABO is motivated specifically by the NIAH setting. As a result, although the performance of ABO is significantly higher than baselines in NIAH, it is not that significant in other practical scenarios. For instance, in Table 3 ABO has similar performance with AT2 in many columns.

---

> ### Author Rebuttal · Authors · 2025-07-31
>
> Thank you for your careful review and encouraging remarks. We have carefully considered your suggestions, and we provide detailed responses and clarifications to each of your comments below.
>
> **Q1. Weakness in Significance.**
>
> **A1.** While ABO is indeed motivated by the NIAH setting as a rigorous stress test, our evaluation goes well beyond this controlled setup. As detailed in Sec. 5.2–5.4, we demonstrate ABO’s effectiveness across a range of real-world applications, including zero-shot toxicity suppression, sentiment steering, and multi-step reasoning error correction.
>
> Regarding Table 3, while ABO and AT2 may appear similar under low pruning ratios (e.g., 10% or 50%), the performance gap becomes substantial under more challenging conditions. For instance, at higher pruning ratios (90%, 95%, 97%), ABO consistently retains significantly more task-relevant information, outperforming AT2 by a wide margin (e.g., up to +15% in accuracy at 95% pruning on QWen3-14B), especially as attribution difficulty increases.
>
> We will make this broader significance more precise in the revised version.
>
>
> **Q2. Why do we measure attention bias instead of original attention value plus attention bias?**
>
> **A2.** Thank you for the insightful question. While both the original attention scores and the optimized bias terms contribute to the model’s output during inference, we deliberately use only the learned attention bias for attribution, motivated by two key reasons below.
> Separation from confounding attention patterns: Original attention scores are distributed across many heads and layers, each with distinct contextual roles and dynamic ranges. These scores often reflect structural or positional biases rather than causal token influence. Including them directly in attribution would risk entangling meaningful signal with these confounding factors.
> Stability and interpretability of the bias space: The learned attention bias vector is optimized end-to-end in a single, unified space. As such, it provides a clean, direct measure of how much “perturbations” (sensitivity or additional emphasis) a token needs to shift the model's output, offering a more faithful and causally grounded saliency signal. Combining it with the original attention would require complex normalization across layers and heads, introducing potential scale mismatches and reducing attribution clarity.
>
>
> **Q3. My concern is that if the attention of a relevant token is already high, the learned bias would turn out to be small, as adding additional bias does not make much impact on the final probability of outputting the target token. Consider the example in Section 4, Line 178, if I set the target token as the first secret message "Yin", what would be the learned attention bias of "Yin"? What would be the original attention of "Yin"?**
>
> **A3.** Thank you for the insightful observation. We agree that a relevant token with already-high attention might require less additional bias to influence the output, and we acknowledge that token position can affect baseline attention. To examine this, we conducted a series of controlled “position rotation” experiments based on the setup in Section 4 (Lines 178–181), where three secret messages (e.g., “Yin,” “Vin,” “Kin”) are inserted into the prompt at fixed positions.
> In the original Figure 5 setting, “Yin” appears first, “Vin” second, and “Kin” third, and the model naturally selects “Yin” as the answer. We apply a positive attention bias only to “Vin” (not the top-1 token), and observe that its generation probability increases step-by-step across decoding steps, eventually overtaking “Yin” and flipping the final output to “Vin.” This shows that even when the target token already receives moderate attention, a learned bias can still exert causal influence and change the model’s behavior.
>
> To test whether this effect is consistent and not due to position-specific attention, we rotate the positions of “Yin,” “Vin,” and “Kin” in all possible orders, and repeat the same procedure: always applying positive attention bias to the message that was not selected by the original model. Across all permutations, we consistently observe the same outcome: the biased token gains generation probability over time, and often replaces the original top-1 answer. This holds regardless of whether the target token is at the beginning, middle, or end of the message list.
>
> These results confirm that the attention bias learned by ABO captures causal influence that goes beyond simply reinforcing tokens with already-high baseline attention. We will include these detailed experimental results and analysis in the revision and Appendix to clarify this key intuition.

---

> > ### Comment · Reviewer_iMDk · 2025-08-02
> >
> > Thank you for your detailed response. It resolves my concerns, and I will keep the original score.

---

### Official Review · Reviewer_6w9t · 2025-07-03

**Clarity:** 3
**Significance:** 2
**Originality:** 3
**Rating:** 4
**Confidence:** 2

**Summary:**

This paper discusses the LLM input saliency methods and proposes the Attention Bias Optimization (ABO) method.
It introduces a systematic evaluation framework based on the needle-in-a-haystack (NIAH) paradigm to stress-test attribution methods, revealing that existing techniques assign substantial importance to irrelevant tokens.
To address this issue, they propose ABO, which frames token attribution as an optimization problem over attention biases, demonstrating superior performance across various tasks, including toxicity suppression, sentiment steering, and reasoning error correction.

**Questions:**

Please refer to the weakness section

- Why share bias parameters across all attention heads and layers?

**Ethical Concerns:**

["NO or VERY MINOR ethics concerns only"]

**Final Justification:**

The authors have addressed my concerns.

**Limitations:**

Yes

**Quality:**

3

**Strengths And Weaknesses:**

**Strengths:**

- The NIAH-based stress testing framework embeds secret messages in distractor text with known ground-truth saliency distributions, which is well-formulated and could become a standard benchmark for future saliency research.

- The proposed method ABO is well-motivated and provides a more direct causal pathway compared to post-hoc gradient-based approaches.

- The paper provides experiments of various models, and the proposed methods show significant improvements in hit ratio. The practical applications (detoxification, sentiment steering, error correction) also indicate applicability in several generation scenarios.

**Weaknesses:**

- The NIAH evaluation primarily tests retrieval-style tasks rather than complex reasoning scenarios where token interactions matter more. The evaluation focuses on relatively simple cases where ground truth is clear-cut, but real-world saliency often involves subtle semantic relationships that may not be captured.

- The paper lacks a theoretical analysis of why attention bias optimization should provide better attribution than existing methods. The connection between attention bias magnitude and true causal contribution is pre-assumed rather than proven. Also, the method requires solving an optimization problem for each input example, which introduces significant computational cost compared to single forward-pass methods.

- The paper should provide more detailed qualitative case studies, e.g. cases when ABO might fail or perform poorly. For instance, how does it handle cases where multiple tokens have legitimate causal relationships?

---

> ### Author Rebuttal · Authors · 2025-07-31
>
> We sincerely thank you for your constructive feedback and appreciate your positive remarks. Below, we provide detailed responses to address your concerns point by point.
>
> **Q1. The NIAH evaluation primarily tests retrieval-style tasks rather than complex reasoning scenarios where token interactions matter more. The evaluation focuses on relatively simple cases where ground truth is clear-cut, but real-world saliency often involves subtle semantic relationships that may not be captured.**
>
> **A1.** Thanks for this comment.
> First, our evaluation is **not limited to retrieval-style scenarios**. As shown in Sections 5.2–5.4, we evaluate ABO across **diverse and semantically rich real-world applications**, including **zero-shot toxicity suppression, sentiment steering**, and **error correction in multi-step reasoning tasks**. These experiments demonstrate that ABO **not only passes basic attribution sanity checks, but also performs robustly** in settings where **token interactions and causal reasoning matter**.
>
> Second, we would like to stress that the **simplicity of the NIAH setup is a deliberate and important design by choice** – it provides a controlled setting with clear ground truth, which allows us to **rigorously sanity-check attribution methods**. In fact, the NIAH test reveals that **many widely used attribution techniques fail even in these simple settings** with **clear-cut** ground truth (see Tables 1 and 2). We believe this limitation has not been found in the literature, and it represents a minimal requirement that attribution methods should meet before applying to more complex scenarios.
>
> **Q2. The paper lacks a theoretical analysis of why attention bias optimization should provide better attribution than existing methods. The connection between attention bias magnitude and true causal contribution is pre-assumed rather than proven.**
>
> **A2.** We appreciate the reviewer’s point! A theoretical analysis would certainly strengthen the work. However, given that our goal is to attribute behavior in billion-parameter LLMs (and potentially larger models), establishing rigorous theoretical guarantees is challenging, if not infeasible. We view this as a limitation of our work, and more broadly, a limitation shared by much of the interpretability literature in the era of LLMs. We will discuss this limitation in the revision.
>
> We also acknowledge the importance of “validate the link between attention bias magnitude and causal contribution”, as mentioned by the reviewer, which is why we included the empirical study at  (Lines 178–209, Figure 5) in Sec. 4. While not a formal proof, this experiment demonstrates that in a controlled and interpretable setting, manipulating attention bias alone can deterministically shift the model’s final output. This serves as a strong empirical justification for using optimized bias magnitude as a proxy for causal influence, and it guided the design of our attribution method.
>
>
> **Q3. Also, the method requires solving an optimization problem for each input example, which introduces significant computational cost compared to single forward-pass methods.**
>
> **A3.** We agree that per-instance optimization introduces higher computational cost than  single forward-pass methods. In fact, we have explored multiple alternative approaches that avoid backpropagation, but found that they consistently underperform in attribution accuracy. This is also evidenced by our experimental results, where ABO substantially outperforms other methods across a range of settings.
>
> However, the computational overhead introduced by ABO is manageable, as it optimizes input-dimension-aligned parameters of size $L$, same as gradient-based saliency methods. For example, in our experiment on the NIAH stress test with a context of 10K, ABO takes approximately less than 2 minutes per data instance on average.
>
>
> **Q4. The paper should provide more detailed qualitative case studies, e.g. cases when ABO might fail or perform poorly. For instance, how does it handle cases where multiple tokens have legitimate causal relationships?**
>
> **A4.** We appreciate the reviewer’s insightful suggestion. Following that, we peered into the failure cases, especially those involving **multiple tokens with legitimate causal relationships.**
>
> Specifically, one challenge we observed arises from **how the learned bias scores are converted into final attribution scores.** The raw attention bias values are unbounded and can be both positive and negative, which makes them unsuitable for direct interpretation. We apply a softmax over the learned biases to produce normalized attribution scores. We used this transformation because of  two main benefits: it maps large negative values to near-zero importance and provides a probability-like interpretation across tokens. However, it also introduces sensitivity, especially in scenarios where several tokens have comparable causal effects. In such cases, the softmax can sharply amplify minor differences in the raw bias values, potentially skewing the final attribution toward one token over others due to random initialization or early optimization dynamics.
>
> In the revision, we will add more detailed qualitative illustrations (e.g., in the style of Figure 3) to demonstrate the failure cases and the lessons that we could learn.  We also see these failure modes not only as limitations, but also as promising directions for future work, for instance, exploring alternative normalization schemes or uncertainty-aware attribution mechanisms.
>
>
> **Q5. Why share bias parameters across all attention heads and layers?**
>
> **A5.** Thank you for this excellent question. We initially experimented with assigning separate learnable attention bias parameters for each layer and head. However, we found that this added complexity did not improve attribution performance. On the contrary, it introduced significant optimization challenges: convergence became slower, and the increased parameter freedom led to instability due to cumulative interactions across layers. In contrast, sharing bias parameters across all heads and layers provided a more stable and lightweight optimization process, while still delivering strong attribution performance. This simplification also reduces the computational and memory overhead, making ABO more practical as a plug-in interpretability tool. We will add this ablation study and discussion in the revision.

---

> > ### Comment · Reviewer_6w9t · 2025-08-05
> >
> > Thank you for clarifying the method. Most of my concerns are addressed. I'll keep the scores.

---

> > > ### Author Response · Authors · 2025-08-05
> > > **Thank you!**
> > >
> > > Dear Reviewer 6w9t,
> > >
> > > Thanks for acknowledging our responses. Your constructive feedback has greatly improved our submission and we will reflect our discussions in the revision.
> > >
> > > Best,
> > >
> > > Authors

---

### Official Review · Reviewer_YGKp · 2025-07-03

**Clarity:** 3
**Significance:** 3
**Originality:** 3
**Rating:** 4
**Confidence:** 4

**Summary:**

This paper proposes a stress-testing framework based on the Needle-in-a-Haystack (NIAH) paradigm, where a hidden "secret message" is embedded in long passages of irrelevant text. Considering seven popular saliency methods, they find: All methods misattribute saliency to irrelevant context, and this error worsens with longer inputs. In prompts over 10K tokens, more than 90% of saliency is incorrectly assigned to distractors. The paper then introduces a method: Attention Bias Optimization (ABO) ABO introduces learnable per-token attention biases into the transformer’s attention logits. It optimizes these biases to maximize the probability of a target output token, with a sparsity inducing regularizer. Saliency is then defined by the magnitude of the learned biases. The paper then shows that effectiveness of this approach across several tasks.

**Questions:**

See the weakness section. But I list them here.

- Model vs. Method Attribution Failures:  To what extent are the attribution failures observed in your stress tests due to limitations of the attribution methods themselves versus the lack of robustness or interpretability in the pretrained LLMs used (e.g., LLaMA2, Mistral)?

- Relevance of Adversarial and Robust Training: Why doesn’t the paper consider or evaluate attribution methods on models that have been adversarially trained or regularized for off-manifold robustness—given prior work shows this leads to significantly better gradient-based attributions?

- Positioning ABO vs. Robust Training: How does ABO compare with or complement models that have been trained explicitly to yield faithful attributions (e.g., via adversarial robustness)? Would ABO still offer performance gains in that setting?

- Clarity on Problem Setting: Is ABO intended as a general replacement for existing saliency methods, or as a solution specifically for frozen, black-box LLMs that cannot be retrained or regularized? This distinction should be made clearer.

- Interaction Between ABO and Model Smoothness: Does ABO implicitly assume or require smoothness in the model’s input-to-output mapping? If so, how sensitive is it to local irregularities (e.g., sharp gradient transitions) in the model's attention landscape?

- Potential for Training-Time Integration: Could ABO be incorporated during training (e.g., as an auxiliary objective) to produce inherently more interpretable models? If so, how might that compare to existing regularization techniques aimed at interpretability?

- Relation to Existing Literature: Why doesn’t the paper more directly engage with the literature on interpretability improvements via robust training? A brief discussion would help clarify how ABO fits into or diverges from this line of work.

**Ethical Concerns:**

["NO or VERY MINOR ethics concerns only"]

**Final Justification:**

The response has satisfactorily addressed all the remaining concerns that I have about this work. In particular, the authors have satisfactorily addressed my concerns around the cause of the improvements by ABO and the engagement with previous work.

**Limitations:**

- There are tasks that sole attention, per token, focus of ABO will miss.

**Paper Formatting Concerns:**

None.

**Quality:**

3

**Strengths And Weaknesses:**

## Strengths

- Clear Problem Statement and Motivation: The paper identifies a limitation in current input saliency methods: misattribution in long-context settings.

- New Evaluation Framework: The adaptation of the Needle-in-a-Haystack (NIAH) framework for saliency assessment is methodologically sound and cleverly designed. The use of both single-token and multi-token ground truths, as well as contradictory secrets, gives a nice testbed for evaluating attribution fidelity.

- Technical Contribution: Attention Bias Optimization (ABO): ABO is a lightweight, and differentiable method for computing token saliency. It leverages the causal relationship between attention bias and model output, moving beyond proxy-based metrics like attention weights or gradients.

- Comprehensive Evaluation: Includes thorough comparisons across methods (7 baselines), LLMs (LLaMA2, Mistral, QWen), and settings (long-context, pruning, toxicity suppression, sentiment control, reasoning correction). ABO shows consistent and significant improvements (20–40% in hit ratios, better resilience under pruning, etc.).

## Weaknesses
This paper proposes a compelling new saliency method, Attention Bias Optimization (ABO), and demonstrates its superiority over several standard attribution techniques. However, in order to contextualize its contributions more rigorously, the paper should directly engage with a body of prior work that challenges the reliability of saliency maps—particularly gradient-based attributions—under specific conditions.

Earlier foundational work (e.g., Hooker et al., 2018 [https://arxiv.org/abs/1806.10758]) showed that standard feature attribution methods can perform no better than random baselines. This result catalyzed a wave of research on more reliable saliency mechanisms. Crucially, subsequent work (e.g., Ilyas et al., 2019 [https://arxiv.org/abs/1905.02175]) argued that these attribution failures are not flaws in the methods per se, but in the models: models learn non-robust features that are incomprehensible to humans but predictive in the training distribution.

More recent work (e.g., Shah et al., 2021 [https://arxiv.org/abs/2102.12781] and Srinivas et al., 2023 [https://arxiv.org/abs/2305.19101]) has shown that regularizing models for adversarial or off-manifold robustness substantially improves the fidelity of gradient-based attributions. These findings imply that attribution quality is tightly coupled to model smoothness and robustness.

With this in mind, the present paper should address several key questions:

- Attribution vs. Model Quality: To what extent are the failures of current saliency methods in the NIAH stress test due to flaws in the methods themselves, versus the fact that the underlying models (e.g., LLaMA, Mistral) were not trained with robustness or interpretability constraints?

- Frozen Model Assumption: The paper assumes that the model is frozen and attribution must operate post hoc. This is a valid and important scenario. However, it would be helpful to acknowledge prior work showing that with proper training, even gradient-based methods can yield reliable attributions. This would clarify that ABO is not necessarily replacing flawed methods, but offering a solution when retraining is infeasible.

- Off-Manifold Robustness: Recent findings suggest that off-manifold robustness is a key factor in obtaining faithful attributions. The authors should discuss how their approach relates to or might benefit from this line of work, e.g, could ABO be combined with robust training to further enhance attribution?

- Clarifying the Problem Setting: There is an important distinction between: Attribution for frozen, pre-trained black-box LLMs, and Attribution for models explicitly trained with interpretability in mind. The paper could benefit from explicitly defining this boundary and clarifying where ABO fits in the broader landscape.

---

> ### Author Rebuttal · Authors · 2025-07-31
>
> Thank you for your thoughtful review and for recognizing the key novelties of our work. Below, we provide detailed responses to the main questions and concerns raised.
>
> **Q1. The paper should directly engage with a body of prior work that challenges the reliability of saliency maps—particularly gradient-based attributions—under specific conditions, such as Hooker et al., 2018, Ilyas et al., 2019, Shah et al., 2021, and Srinivas et al., 2023.**
>
> **A1.** We thank the reviewer for pointing out these important references. We have added citations to Hooker et al. (2018), Shah et al. (2021), and Srinivas et al. (2023) in the revised version, and we appreciate your suggestion to connect their findings to our work.
>
> The reason we did not initially include these papers is that they primarily focus on saliency methods in the pre-LLM era, particularly in the context of discriminative models such as image classifiers. As a result, we felt their scope was somewhat far from  our focus on token attribution in LLMs. That said, we agree that the fundamental concerns they raise about saliency reliability are highly relevant, and we are glad to incorporate these discussions into our related work, problem scope and setup, and future work.
>
> **Q2. Attribution vs. Model Quality?**
>
> **A2.** This is a great question.
> First, to isolate the effects of the attribution methods from the capabilities of the tested models, we ensured that the LLM produces the **correct answer to every NIAH query before applying any saliency evaluation.** This guarantees that the attribution scores are tested only on examples where the model behavior is correct, thereby attributing performance differences solely to the attribution methods themselves. This design choice is also reflected in Tables 1 and 2: for instance, we evaluate LLaMA2-7B up to 4K tokens and Mistral-7B up to 10K. The reason for this discrepancy is that **LLaMA2-7B fails to consistently retrieve the correct message beyond 4K tokens,** and thus we exclude longer contexts for that model. In short, the attribution failures observed in our study are **not due to model misbehavior,** but rather stem from **systematic shortcomings in the attribution methods.**
>
> Second, we agree that evaluating attribution methods on robustness- or interpretability-enhanced models is a promising direction. However, the goal of our work is to improve the fidelity of attribution mechanisms for general-purpose LLMs without relying on specialized training. Assuming the availability of interpretability-optimized models would limit the applicability of attribution tools in real-world settings, where such models are rarely available. While prior works (e.g., Srinivas et al., 2023) have suggested training-time interventions for interpretability in vision models, extending such techniques to LLMs poses substantial challenges. The computational burden of LLMs renders robust training methods such as the simplest randomized smoothing or the min-max optimization-based adversarial training computationally-prohibitive in practice. We will further discuss this in our response to Q4.
>
>
> **Q3. Frozen Model Assumption.**
>
> **A3.** We fully agree with the reviewer. Our goal is not to claim that ABO should replace all existing methods, especially when models are trained with interpretability constraints. Instead, our work aims to highlight that **current evaluation protocols may give overly optimistic or misleading impressions** of attribution quality when using standard general-purpose LLMs, especially in long-context settings. By introducing the **NIAH** stress test, we hope to **encourage more rigorous and ground-truth-based evaluation**. ABO offers a practical and more effective solution in such cases, and we view it as an improved **complement to, rather than a replacement for,** existing attribution methods.
>
> **Q4. Off-Manifold Robustness.**
>
> **A4.** We appreciate the reviewer’s insightful suggestion. While off-manifold robustness is indeed relevant to attribution, most existing work in this area focuses on image classifiers , not LLMs. In contrast, adversarial training for LLMs presents substantial challenges due to the discrete nature of text inputs, the sheer scale of modern models, and the risk of degrading core generation quality.
> Yet, motivated by the reviewer’s insightful comment, we conducted a detailed survey of recent advances in adversarial training for LLMs. As a result, we carried out **additional experiments** testing ABO on adversarially trained LLMs from the NeurIPS 2024 paper “Efficient Adversarial Training in LLMs with Continuous Attacks” (e.g., Phi-CAT, Zephyr-CAT). Unfortunately, these models did not satisfy the basic requirement for our attribution evaluation: achieving correct retrieval in the NIAH test. Specifically, they attained <15% accuracy even at 1K-token context length. This failure suggests a potential tradeoff between adversarial robustness and generative accuracy, which makes the studied adversarially trained LLMs improper for saliency evaluation.
> Nonetheless, we still view the intersection of attribution and robust training as a promising direction in LLMs. We envision that ABO and robust training approaches can be complementary: future work could explore whether integrating attention bias optimization with robustness-enhancing objectives might yield even more faithful attributions.
>
>
> **Q5. Clarifying the Problem Setting.**
>
> **A5.** Thank you for raising this distinction. We would like to clarify that **ABO does not require the underlying LLM to be explicitly trained with interpretability in mind**, in fact, such LLMS are currently rare in practice. Our goal is to develop a **practical and precise attribution method** that can operate on standard pre-trained (or frozen) LLMs without requiring any model modifications. Like many saliency-based attribution methods, ABO is also **designed for post-hoc, white-box settings**, where we have access to the model’s internal attention mechanism.
>
> As we responded to Q3, we view ABO as an improved **complement to, rather than a replacement for,** existing attribution methods. We will make this distinction clearer in the revised version.
>
>
> **Q6. Interaction Between ABO and Model Smoothness?**
>
> **A6.** As we responded to Q2, our method does **not explicitly assume or require smoothness** in the model’s input-to-output mapping. While we agree that sharp local irregularities (e.g., abrupt gradient transitions) could, in theory, pose challenges for optimization-based attribution, our empirical results suggest that **ABO remains robust and stable across a wide range of applications**, including NIAH stress test (Sec. 5.1), toxicity suppression (5.2), sentiment steering (5.3), and LLM error correction (5.4). These experiments collectively span various data modalities, context lengths, and attribution goals. The consistently strong performance of ABO across all these settings suggests that it does not suffer from instabilities caused by lack of smoothness in practical use cases.
>
> **Q7. Potential for Training-Time Integration.**
>
> **A7.** We appreciate the reviewer’s thought-provoking question. However, we are conservative to incorporating ABO into LLM training as an auxiliary objective. This is because in the era of large-scale language models, modifying the training process for a specific downstream property, such as interpretability, may **compromise core, general-purpose model capabilities**, as also seen in adversarially trained models (see our response to Q4). While integrating interpretability-aware training objectives with ABO is an interesting direction, it requires careful investigation beyond the scope of this work. We will discuss this in our conclusion for future research plans.
>
>
> **Q8. Relation to Existing Literature.**
>
> **A8.** Thank you for the suggestion. We will enrich our literature review per suggestion. We also agree that robust training is an important line of work in interpretability research. However, as we responded to Q1, Q2, Q4, and Q5, our scope is centered on **LLMs**, where robust training techniques are **not widely adopted for LLMs** and may  **degrade core generation capabilities.**

---

### Comment · Area_Chair_tJnX · 2025-08-07
**Please participate in discussions with authors before submitting “Mandatory Acknowledgement”**

Dear Reviewers,

I see that some of you have not yet responded to the author's rebuttal with further feedback. Please make sure to do so, which is important to reach a well thought out  consensus for the quality of this paper.

Best regards,
Submission18179 AC

---

### Decision · Program_Chairs · 2025-09-17

**Decision:**

Accept (spotlight)

**Comment:**

This paper proposes a stress-testing framework based on the Needle-in-a-Haystack (NIAH) paradigm, where a hidden "secret message" is embedded in long passages of irrelevant text.   The authors find existing saliency methods misattribute saliency to irrelevant context, and this error worsens with longer inputs. Based on this observation a new Attribution method is proposed based on  Attention Bias Optimization (ABO), which optimizes these biases to maximize the probability of a target output token, with a sparsity inducing regularizer. The effectiveness of this approach is demonstrated across several realistic tasks  (detoxification, sentiment steering, error correction) .

This work represents a solid improvement over a family of important and realistic tasks. However, the benchmark and solution are limited to NIAH settings rather than more complex reasoning scenarios. Also the result will be more complete if the relationship to related work can be explained in more detail (e.g., Attribution vs. Model Quality, Frozen Model vd fine tuned, Off-Manifold Robustness).